# Community Knowledge about Autism Spectrum Disorder in the Kingdom of Saudi Arabia

**DOI:** 10.3390/ijerph19063438

**Published:** 2022-03-14

**Authors:** Amal Khaleel Abualhommos, Abdullah Hamad Aldoukhi, Ammar Ali Abdullah Alyaseen, Fatima Ali AlQanbar, Naimah Alshawarib, Zainab Abbas Almuhanna

**Affiliations:** Pharmacy Practice Department, Clinical Pharmacy College, King Faisal University, Alhasa 43518, Saudi Arabia; aldoukhi1996@gmail.com (A.H.A.); al-yaseen1997@hotmail.com (A.A.A.A.); fatiimaaqa@gmail.com (F.A.A.); n3oomah.5@hotmail.com (N.A.); zainb3bbas@gmail.com (Z.A.A.)

**Keywords:** autism, knowledge, Saudi Arabia

## Abstract

Objectives: To explore the knowledge of the general community in Saudi Arabia about autism spectrum disorder. Method: A cross-sectional study was conducted for the period between June and September 2021 in Saudi Arabia using an online questionnaire tool. The questionnaire tool was developed based on a literature review. The questionnaire tool consists of 34 items that assess knowledge about autism spectrum disorder in terms of its etiology, autistic patient features, autistic children’s abilities and needs, and autistic adults’ abilities and needs. The total score for each subscale was used to define the level of knowledge of it. Correct answers were given a score of one, and the total score for each subscale was used to describe the level of knowledge of it. Logistic regression was used to identify predictors of good knowledge about autism spectrum disorder (defined as a total score equal or above the mean score of the study participants). Results: This study enlisted the participation of 500 people. The participants’ overall understanding of autism spectrum disorder was moderate, with a mean score of 20.6 (SD: 5.6) out of 34, or 60.6%. The participants’ knowledge levels ranged from 32.2% to 77.5%. The items about the abilities and needs of adolescents and young people with autism had the highest degree of knowledge (77.5%). The items about autism’s causes had the lowest level of expertise (32.2%). When compared to others, females, those with a master’s degree, and those working in the healthcare field had a higher likelihood of knowing more about the autism spectrum condition. (*p* ≤ 0.05). Conclusion: Knowledge about autism spectrum disorder in Saudi Arabia is moderate. Social media channels and healthcare centers should be used to conduct educational campaigns for parents. The goal of this educational campaign should be to improve parents’ ability to recognize the causes of autism.

## 1. Introduction

ASDs (Autism Spectrum Disorders) are a type of neurodevelopmental disorder that last a lifetime [1]. This impairment results in functional and structural alterations in the brain, reducing sociability and communication with others, as well as a repeating patterns of sensory and motor behavior. There is typically nothing about people with ASD that distinguishes them from others, but they require a unique approach to learning how to behave and communicate. Although some ASD patients could have a mild condition that requires minor caregiver support, others may have a severe case that requires a significant amount of support [1]. Autism can present differently from person to person, despite the fact that certain features are common across individuals/children with ASD.

In the last two decades, the incidence (number of new cases) has increased. This disorder affects one out of every 44 children aged eight to sixteen, with boys having a 4.2-fold higher frequency than girls [2,3]. Increased public knowledge of the autistic disease is one cause for this increase; as a result of this awareness, more parents are seeking screening for their children’s problems, resulting in medical confirmation of the diagnosis [4].

The American Psychiatric Association’s Diagnostic (DSM-5) criteria were published in 2013 and replaced the previous version, which had different subtypes of ASD such as autistic disorder, pervasive developmental disorder, and Asperger’s disorder (DSM-IV). Instead of having several subcategories of ASD such as autistic disorder, pervasive developmental disorder, and Asperger’s disorder, it was proposed that we have only one ASD diagnosis based on issues in two basic domains: social communication and confined and repetitive sensory activities [5].

A child must show persistent deficiencies in each of three domains of social communication and interaction, as well as at least two of four types of restricted, repetitive behaviors, to meet the DSM-5 diagnostic criteria for ASD [1]. Autism in Toddlers and Young Children (STAT) and Autism Diagnostic Observation Schedule are two further screening instruments (ADOS) [6]. One of the most often utilized screening methods for autism diagnosis is the Modified Checklist for Autism in Toddlers (M-CHAT) [7].

ASD can be diagnosed at 1.5 years [8], however, the mean age for the diagnosis is 3.3 years [9]. In comparison to a child who has no sibling, if the child has an older sibling, he will receive an earlier diagnosis [10]. ASD is difficult to diagnose because there is no medical test, such as a blood test, or a reliable biomarker, to confirm the disorder. However, there are some early symptoms of the disorder based on their behavior, such as avoiding direct eye contact with their caregiver and a lack of interest in playing with their friends; they also have limited vocabulary and a set daily routine [1].

Early detection of the disease leads to better outcomes, such as enhancing the child’s verbal, cognitive, and communication abilities, as well as their problem-solving activities and physical development [3]. In Saudi Arabia, even after confirming the diagnosis, treatment may be delayed due to the parent’s educational level and understanding of ASDs, as well as their annual income and geographic location in the country, all of which introduce different considerations with regard to commencing treatment [11].

Attention-deficit hyperactivity disorder (ADHD), social anxiety disorder, intellectual disability, and epilepsy are all examples of developmental disabilities that can coexist with concomitant mental diseases and a separate genetic disorder [12]. ADHD is another neurodevelopmental disease that frequently coexists with ASD. It affects both children and adults and can lead to depression and anxiety [13]. The intellectual level of ASD children will vary widely, with some having a severe disability and others possibly outperforming their healthy peers [14].

Although the exact cause of ASD is unknown, it is believed that environmental, lifestyle, and genetic risk factors such as maternal age older than 40 years old, excess body weight, hypertension, bacterial or viral infection during pregnancy, and the use of medications such as valproic acid all play a role [15,16]. Besides, If the child has a sibling with ASD, his or her chances of developing ASD increase by 7–20% [17,18]. According to the World Health Organization, there is no evidence of a link between childhood vaccines and ASD [19], including measles, mumps, and rubella (MMR) nor any other childhood vaccine [20].

As there is no one cure for ASD, each child requires tailored therapy based on his or her specific condition, and parents must also participate in skill training programs to improve the quality of life of both the ASD child and themselves [20]. Besides, the cost of treating ASD is relatively high, and it is even higher for children with severe intellectual disabilities, putting a direct financial pressure on their caregivers, particularly their mothers [21,22]. According to a previous review study, the prevalence of ASD in Arab countries is higher than in other developing countries, with 42,500 verified cases in 2002 and possibly many more undiagnosed cases [23]. The prevalence in Arab Gulf countries ranges from 1.4 to 29 individuals per 10,000 people [24]. According to a previous study in Saudi Arabia, the prevalence of ASD in two major cities in the western area of the country (Makkah and Jeddah) was reported to be 2.81 per 1000 people [25]. Research on ASD in different regions and cities within the Kingdom of Saudi Arabia is limited; even in the country’s main cities, little is known about the community knowledge about ASD [23]. According to the previous information, this study aims to assess the community knowledge about ASD in the Kingdom of Saudi Arabia. Exploring community knowledge of ASD is crucial because it contributes significantly to requesting screening for children with ASD, which leads to medical confirmation of the diagnosis and the initiation of treatment at an earlier stage.

## 2. Method

### 2.1. Study Design

A cross-sectional study using an online survey was conducted in Saudi Arabia between June and September 2021.

### 2.2. Study Population and Sampling Procedure

All individuals who are currently living in Saudi Arabia and aged 18 years and above formed the study population. The questionnaire was distributed through social media platforms. A convenience sampling technique was used to recruit the study sample. The link was distributed using social media platforms (WhatsApp and Facebook). The study aims and inclusion criteria were mentioned in the cover letter.

### 2.3. The Questionnaire Tool

The questionnaire tool was developed based on a literature review [1,26,27]. The questionnaire tool was written in Arabic to ensure that the majority of Saudi Arabia’s population could understand it. The questionnaire tool was comprised of 34-items that measure the level of knowledge about autism spectrum disorder in term of its causes (nine -items), characteristics of autistic patients (9-items), the abilities and needs of children with autism (8-items), and the abilities and needs of adults with autism (8-items). Each correct response counts for 1 point, and 0 points for incorrect responses. The higher the points, the more knowledgeable the participants. As the knowledge section was comprised of 34 questions, the maximum obtainable score was expected to be 34.

The clarity of the questionnaire items and whether they are measuring people’s knowledge of ASD were discussed with two professional clinicians. They validated this and gave comments on some of the wording used in presenting the questionnaire items in order to make them more understandable to the general audience. Before disseminating the final questionnaire to the research population, all comments were addressed.

Pilot research was done on 25 people from the general public utilizing the developed questionnaire tool. Participants were asked about the questionnaire’s clarity and comprehensibility, as well as whether any of the questions were difficult to comprehend. They confirmed that the questionnaire was simple to comprehend and fill out.

### 2.4. Statistical Analysis

Descriptive statistics were used to describe participants’ demographic characteristics. Continuous data were reported as mean ± SD. Categorical data were reported as percentages (frequencies). Logistic regression was used to identify predictors of good knowledge about autism spectrum disorder (defined as a total score equal or above the mean score of the study participants, which is 20.6). A two-sided *p* < 0.05 was considered statistically significant. The statistical analyses were carried out using SPSS (version 25) (SPSS Inc., Chicago, IL, USA).

## 3. Results

### 3.1. Participants Demographic Characteristics

A total of 500 participants were involved in this study. Around 47.0% of them were aged 18–25 years. More than half of them (76.4%) were females. A total of 64.6% of them had a Bachelor’s Degree. One third the study participants (33.0%) were working in the healthcare sector. Table 1 describes the demographic characteristics of the study participants.

### 3.2. Knowledge about Autism Spectrum Disorder

The vast majority of the study participants (80.4%) reported that they had prior knowledge about autism spectrum disorder. The main source of their information about it was social media (57.2%). The vast majority of the study participants (80.4%) reported that they had moderate to high levels of knowledge about this disorder, Table 2.

Participants showed variation regarding their knowledge about autism spectrum disorder. The highest knowledge score was found for items related to knowledge of the abilities and needs of adolescents/young people and children with autism disorder and characteristics of children with autism spectrum disorder (77.5%,70.0%, and 65.6%). However, the level of knowledge regarding the causes of this disorder was low (32.2%), Table 3.

Figure 1, Figure 2, Figure 3 and Figure 4 below describe participants’ responses to each item that measure their knowledge about autism spectrum disorder.

### 3.3. Participants Knowledge Score and Their Predictors

Binary logistic regression analysis was used to identify predictors of good knowledge about autism spectrum disorder, Table 4. Males and those who work in non-medical fields had lower odds of being knowledgeable about autism spectrum disorder compared to others with (OR: 0.58 (95%CI 0.38–0.88) and (OR: 0.53 (95%CI 0.37–0.77)), respectively. On the other hand, participants with a Master’s Degree and those who work in the healthcare field showed higher odds of being knowledgeable about autism spectrum disorder compared to others with (OR: 3.55 (95%CI 1.01–12.41)) and (OR: 1.86 (95%CI 1.25–2.75)), respectively.

## 4. Discussion

The aim of this study was to assess the community knowledge about ASD in the Kingdom of Saudi Arabia. The key findings are: (1) the vast majority of the study participants reported that they had prior knowledge about autism spectrum disorder, (2) the main source of their information about it was social media, (3) the vast majority of the study participants reported they have moderate to high levels of knowledge about this disorder, (4) the highest knowledge score was found for items related to knowledge of the abilities and needs of adolescents/young people and children with autism disorder and characteristics of children with autism spectrum disorder, (5) the level of knowledge regarding causes of this disorder was low, (6) males and those who work in non-medical fields were less likely to be knowledgeable about autism spectrum disorder compared to others, and (7) participants with a Master’s Degree and those who work in the healthcare field were more likely to be knowledgeable about autism spectrum disorder compared to others.

These findings are in contrast to a previous study [28] which found that not only the general population, but even certain healthcare providers and educators were unfamiliar with autism. In our study, the vast majority of participants (80.4%) reported that they had prior knowledge of ASD. The majority of them had moderate to high awareness of autistic children’s characteristics and abilities, but only a minimal understanding of the disease’s causes. These findings show a high level of knowledge when compared to a prior survey conducted in Saudi Arabia, which found that the degree of understanding was only 41% [29]. One study in Australia and another in Africa found similar results to ours; in both studies, participants had a general understanding of ASD but very little understanding of the disease’s cause [30,31]. The majority of Australians were aware of ASD, but they believed it was caused by the MMR vaccine. However, the World Health Organization (WHO) stated that there was no evidence linking any vaccine to autism [19,30]. Another study revealed a misunderstanding about the treatment, with participants believing that children with ASD are unable to attend public school [31]. As previously said, this disorder is caused by a number of variables, some of which are genetic in nature and others which are environmental in nature [15,16,17,18].

In our study, participants showed variation regarding their knowledge about autism spectrum disorder. The highest knowledge score was found for items related to knowledge of the abilities and needs of adolescents/young people and children with autism disorder and characteristics of children with autism spectrum disorder. However, the level of knowledge regarding causes of this disorder was low. ASD is caused by a combination of genetic and environmental factors. Knowing ASD risk factors is critical for reducing the likelihood of having children with ASD [1,32].

In our study, social media was the primary source of participant information about ASD. In the fast changing Saudi society, social media has become increasingly important. Saudi Arabia is a social media powerhouse, with one of the world’s largest national markets for Snapchat and YouTube. In 2021, it was anticipated that 95.7% of the population, or roughly 35 million people, were active internet users. The majority of them (79.2%) are social media users who spend an average of 3 h every day using it [33], with more than 25 million active users for Facebook.

Social media has a huge impact on our lives and knowledge nowadays. Some films serve as both entertaining and educational tools. People rely on TV shows and movies to learn about diseases such as autism [34]. One study, for example, looked at the impact of 23 different ASD Hollywood films on the general public, medical students, and psychiatric trainees, and found that they improve public awareness as well as medical education [35]. In another study, they analyzed ASD characters in 15 different films using an assessment tool like the Childhood Autism Rating Scale (CARS2) and compared their symptoms to the usual distribution of true symptomology [36]. People rely on TV shows and movies to learn about autism [34]. At the same time, there is growing concern that these social media platforms are also disseminating inaccurate or misleading health information [37]. As a result, it should not be used as a sole source of health-related data.

Participants in our study who worked in the health-care field and had a high level of education had a greater knowledge and understanding of ASD than those who did not (OR: 1.86 (95%CI 1.25–2.75)). On the other hand, a previous study conducted in Saudi Arabia revealed a moderate to poor degree of ASD knowledge across various HCPs, with a mean average of 51.5% [38].

In our study, participants with a Master’s Degree or higher were 3.5 times more likely to be knowledgeable about autism spectrum disorder compared to others. This was confirming the findings of a previous study which found that those with a higher education had a better attitude, but those with prior experience with people with ASD had the best results [39]. In addition, in our study, males and non-medical sector professionals had the least understanding of ASD, which was confirming the findings of previous studies [3,15,29,30,39,40]. Males in one study believed that ASD is not a lifelong condition and that it can be cured. These ideas cause parents of autistic children to feel like they have failed to support their child, and they begin to reject their child [30]. In another study, women had a much better attitude toward ASD than males, with a difference of 6% [40]. Another study also found that women have a more positive attitude toward ASD, as well as schizophrenia and bipolar disorder [40]. Besides, having prior experience with ASD people was another important factor that influenced the attitude positively [3].

Better outcomes can be attained if the condition is detected early [30], potentially causing dramatic changes in a child’s linguistic, cognitive, and adaptive behavior [41,42]. Furthermore, before the child reaches the age of four, early intervention will improve the child’s social behavior and daily skills [43], keeping in mind that early detection is not always a simple process that confirms the diagnosis [4], since it is highly dependent on the level of education of the HCP and the execution of ASD guidelines [4]. According to a prior study, the time between the initial observation of the child’s behavior and the confirmed diagnosis could be up to two years [44]. Families with ASD children will experience more stress as a result of late diagnosis and intervention [45]. When a parent’s child is diagnosed with ASD, it has a significant impact on the entire family [44,46]. According to one study, 80% of ASD children have a sleep problem, which has an impact on their entire family’s resilience when compared to parents of ASD children who do not have a sleep problem [47]. Furthermore, having an ASD child adds to the family’s stress, resulting in increased levels of depression, anxiety, and stress [48]. When parents of children with ASD and parents of children with Down Syndrome (DS) were compared, it was observed that the ASD group was more stressed [49].

A variety of approaches could be used to enhance awareness of ASD, aid in early diagnosis, and improve patient outcomes. Social media campaigns could be one effective approach to enhance the community awareness and knowledge about ASD. Having a support group for the parents is one approach to assist them [44]. Another option is to establish a positive working connection with a professional HCP [50]. Furthermore, enhancing the degree of ASD parent education decreased stress, increased quality of life, and decreased social isolation [51].

This study does have some limitations. The cross-sectional study design does not allow for causality to be established between study variables. Because this study used a nonprobability sample technique, convenience sampling may have influenced the generalisability of our findings. Data collection via an online survey may have missed some of the intended demographic groups. However, this was standard research practice during the pandemic, as a recent study found that the usage of online evaluation platforms grew dramatically during the COVID-19 epidemic, particularly for educational and research purposes [52]. Finally, our newly constructed questionnaire instrument (not standardized) was tested on a small group of people from the general public with no further validation procedures.

Future research should look into other community sectors with poor ASD awareness and find effective educational interventions to raise community awareness and knowledge. Policymakers in the healthcare sector should improve parents’ understanding of various developmental disabilities and promote early detection through frequent educational campaigns targeted at people at high risk of having children with ASD.5.

## 5. Conclusions

According to this survey, the Saudi community has a moderate to high level of knowledge regarding ASD. The public’s understanding of the disorder’s causes is limited. There is a need for an educational campaign that uses a variety of social media platforms. Healthcare practitioners are also requested to emphasize the importance of improving parents’ capacities and awareness of ASD’s causes and symptoms.

## Figures and Tables

**Figure 1 ijerph-19-03438-f001:**
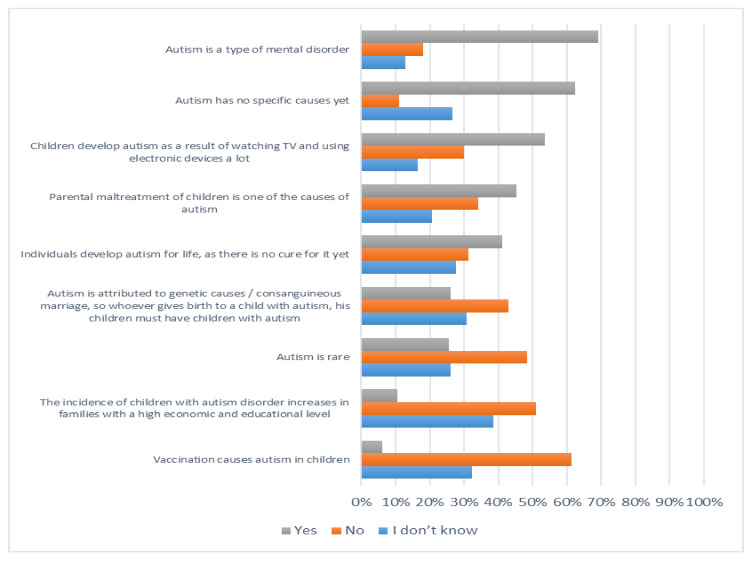
The level of knowledge of the causes of autism disorder.

**Figure 2 ijerph-19-03438-f002:**
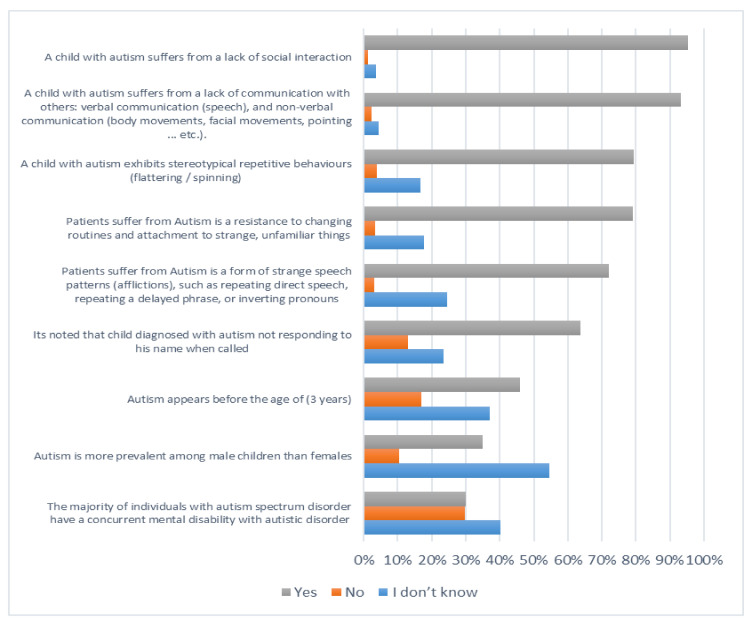
The level of knowledge of the characteristics of children with autism disorder.

**Figure 3 ijerph-19-03438-f003:**
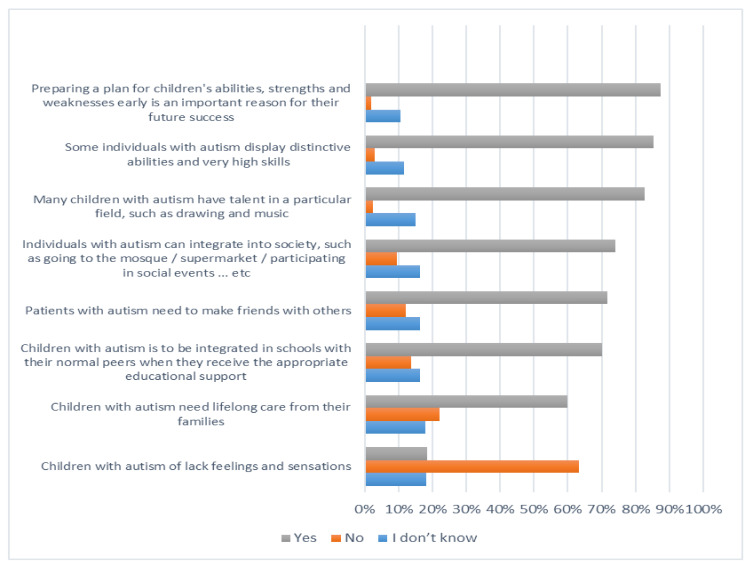
The level of knowledge of the abilities and needs of children with autism disorder.

**Figure 4 ijerph-19-03438-f004:**
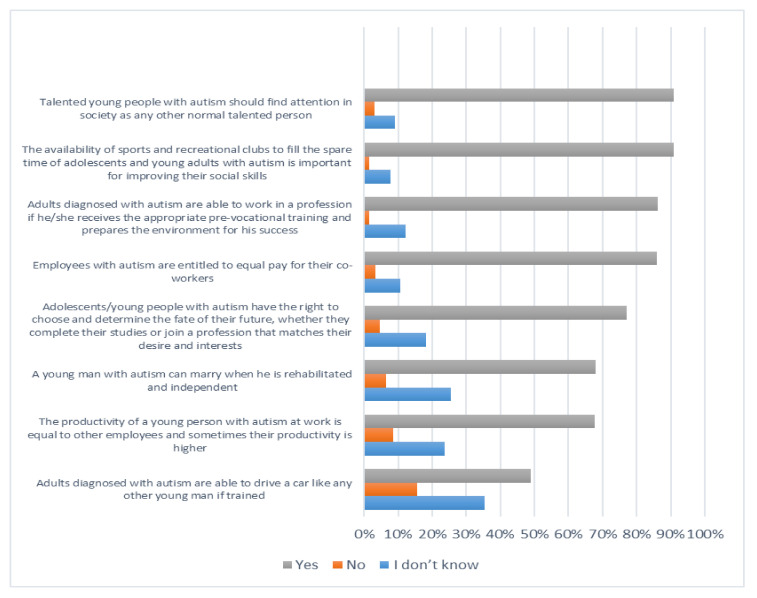
The level of knowledge of the abilities and needs of adolescents/young people with autism disorder.

**Table 1 ijerph-19-03438-t001:** Participants demographic characteristics.

Demographic Variable	Frequency	Percentage
Age		
18–25 years	233	46.6%
26–35 years	104	20.8%
36–45 years	86	17.2%
46 years and above	77	15.4%
Gender		
Females	382	76.4%
Education level		
Secondary school level or lower	146	29.2%
Bachelor’s degree level	323	64.6%
Master’s degree level	18	3.6%
PhD level	13	2.6%
Employment status		
Unemployed	105	21.0%
Student	25	5.0%
Working in the healthcare sector	165	33.0%
Working outside the healthcare sector	194	38.8%
Retired	11	2.2%

**Table 2 ijerph-19-03438-t002:** Knowledge about autism spectrum disorder.

Variable	Frequency	Percentage
I have prior knowledge about autism spectrum disorder:		
Yes	402	80.4%
If yes, information Sources about Autism Spectrum Disorder: (more than one answer could be selected)		
Social media platforms	286	57.2%
Books and scientific articles	127	25.4%
Internet	121	24.2%
Friends and family	80	16.0%
The level of knowledge of the causes of autism disorder		
How do you describe your level of knowledge of the causes of autism disorder?		
I don’t know anything	98	19.6%
I have moderate level of knowledge	220	44.0%
I have high level of knowledge	182	36.4%

**Table 3 ijerph-19-03438-t003:** Participants knowledge score.

Scale	Number of Items	Mean (SD)	Percentage of 100% from Total Score of the Scale
Knowledge about causes of autism	9	2.9 (1.4)	32.2%
Knowledge about characteristics of children	9	5.9 (1.9)	65.6%
Knowledge about the abilities and needs of children with autism	8	5.6 (1.9)	70.0%
Knowledge about the abilities and needs of adolescents/young people with autism	8	6.2 (2.1)	77.5%
Total score for all items	34	20.6 (5.6)	60.6%

**Table 4 ijerph-19-03438-t004:** Logistic regression analysis.

Demographic Variable	Odds Ratio (95%CI)	*p*-Value
Age		
18–25 years (Reference group)	1.00	
26–35 years	1.06 (0.68–1.65)	0.784
36–45 years	1.26 (0.78–2.04)	0.345
46 years and above	0.66 (0.40–1.07)	0.091
Gender		
Females (Reference group)	1.00	
Males	0.58 (0.38–0.88)	0.010 *
Education level		
Secondary school level or lower (Reference group)	1.00	
Bachelor’s degree level	0.97 (0.67–1.41)	0.870
Master’s degree level	3.55 (1.01–12.41)	0.048 *
PhD level	1.10 (0.35–3.40)	0.874
Employment status		
Unemployed (Reference group)	1.00	
Student	0.86 (0.38–1.94)	0.723
Working in the healthcare sector	1.86 (1.25–2.75)	0.002 **
Working outside the healthcare sector	0.53 (0.37–0.77)	0.001 **
Retired	0.56 (0.17–1.87)	0.347

* *p* ≤ 0.05; ** *p* ≤ 0.01.

## Data Availability

Not applicable.

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
