# Peer review of "Community Knowledge about Autism Spectrum Disorder in the Kingdom of Saudi Arabia"

_ijerph, 2022, doi:10.3390/ijerph19063438_

Round 1
Reviewer 1 Report
Please see attached file.

Author Response
Thank you for the opportunity to review the manuscript, “Community Knowledge about Autism Spectrum Disorder in the Kingdom of Saudi Arabia.”
The authors conducted a study to assess public knowledge about ASD. The paper was interesting and aligns well with the journal’s focus. However, there are issues with the study that should be addressed before publication.
Please see my suggestions below.
General • Please check English phrasing throughout.
Introduction:
- Emphasize all individuals/children with ASD are different. Although certain characteristics are common, autism can present differently from person to person.
- Thank you for this comment, we have now addressed it in page 1-2, lines 40-42.
- What is meant by “most children with ASD appear normal” (p. 1, line 36)? Clarify.
- We have now clarified the sentence to highlight that there is typically nothing about people with ASD that distinguishes them from others, but they require a unique approach to learning how to behave and communicate, lines 37-38.
- Page 1, lines 40-41- Awkward. Rephrase.
- We have now addressed the above mentioned comment.
- Page 2, lines 70-75. Misinformation. ADHD, anxiety, ID, and epilepsy are not “ASDs” – they are developmental disabilities.
- We have now addressed the above mentioned comment, line 91.
- The introduction provides an overview of ASD, but does not set up the need for the study. Why do we need to know about public knowledge of ASD in the Kingdom of Saudi Arabia?
- We have now addressed the reviewer comment and highlighted the need for such research in Saudi Arabia, lines 112-124.
Method
- Sections 2.2 and 2.3 are repetitive. Combine.
- We have now addressed the above mentioned comment, lines 129-136.
- Page 3, line 105 – provide examples of the literature the questionnaire was based on.
- We have now addressed the above mentioned comment, line 138.
- Why did the researchers use a logistic regression? The score on the questionnaire provides a continuous variable, yet the researchers selected a random cut-off for a “good score.” The analyses would be more methodologically sound and provide more detailed information if the score was kept as continuous and the authors used a linear regression.
- We would like to thank the reviewer for this valuable comment. Actually, the use of cut-off point to define “good score” was based on the mean knowledge score of our study sample, which is the best measure to present the central tendency of the knowledge score. This procedure of using the mean score of the study sample is frequently utilised across different research area using the mean value of the outcome of interest across the study sample.
Results
- Provide a mean age for participants.
- Unfortunately, we did not collect the age as a continuous variable but as categorical variable (we provided the participants four age group categories to choose between them), so we are not able to provide the mean value.
- Remove horizontal lines on data tables.
- The layout of the tables was formatted by the journal editorial team member. We can’t change it as we need to stick to the formatting guidelines of the journal.
- See above comment regarding regression analysis. Discussion
- As we have mentioned above we used cut-off point to define “good score” based on the mean knowledge score of our study sample, which is the best measure to present the central tendency of the knowledge score. This procedure of using the mean score of the study sample is frequently utilised across different research area using the mean value of the outcome of interest across the study sample. Using the mean score enabled us to build up our dummy variable for the logistic regression model and identify predictors of good knowledge about ASD.
- Please check English throughout.
- We have now addressed the above mentioned comment.
- Page 7, line 179 – awkward, revise sentence.
- We have now addressed the above mentioned comment, line 221.
- Page 8, lines 202-217 – this paragraph is important to understanding your findings, but your message gets muddled. Take time to flush out your findings in the context of other studies here.
- Thank you for this valuable comment. Based on the reviewer comment, we have now rewrote the above mentioned paragraphs to make them clearer for the reader.
- Page 8, lines 219-233 – this paragraph needs a better set up. It does not flow from the prior paragraph.
- We have now addressed this comment.
- Page 8, lines 234-238 – Since most participants were getting their info from social media, perhaps suggest a social media campaign aligned with educating people in the Kingdom of Saudi Arabia about ASD.
- We have now addressed this comment, lines 324-325.
- Discussion should include limitations, implications, and future research.
- We have now addressed this comment, lines 330-343.
Reviewer 2 Report
I would like to thank the authors for submitting an interesting and timely manuscript for review. I found it interesting to learn about the differences in awareness in this specific population and I think this article will be of interest to the journal's general readership. I did have the following comments/suggestions:
- Introduction: In general I found the Introduction informative however it is missing some perspective on the actual burden of ASD in Saudi Arabia (ie: incidence/prevalence if known). I also think the introduction would benefit from more information on Saudi Arabia itself - for example the general health literacy of citizens, what kinds of social media platforms are most utilized etc. If there is a low rate of ASD diagnoses in the kingdom for example, then the study would have much lower significance. In addition to this, what percentage of the population has access to social media platforms (if known)? Obviously, if only a well-educated portion of the population use these, this will skew data accordingly.
- Methods: I would suggest listing ALL platforms on which the survey was advertised if possible. Also was the survey available in English, Arabic, both or any additional languages? Once again if only available in a single language, this will also skew data towards those with only a working knowledge of that language.
- Methods: My main concerns with the article are with the methods - specifically the questionnaire itself. Was the questionnaire validated in any way? Did the authors utilize a pilot group first before releasing the final version to the public? Also on Page 3 Line 105 the authors mention that the tool was developed based on "literature review". If that is the case, the authors should cite some of the pertinent literature (particularly if the current questionnaire did not undergo a rigorous validation process). If the survey/questionnaire was not validated, that should be mentioned as a limitation in the discussion.
- Discussion: In general, I found the discussion lacking in terms of conclusions from the authors. On Page 8 Line 192 for example, the authors comment on social media being the primary source of participant knowledge - yet the authors do not attempt to explain why this may be so in Saudi Arabia specifically. What is it about that population that may make this so?
- Discussion: As well on Page 8 in Paragraphs 3 and 4 the authors spend a lot of time discussing previous studies but do not give any indications on what their study may or may not tell us about the population of Saudi Arabia as they see ASD. Only the final Paragraph 5 suggests a few interventions but these are on briefly mentioned. There is also no mention of the limitations of the current study or how it may be improved in the future. As mentioned, the discussion needs to be modified to really address what the authors have gleaned from their study.
Author Response
I would like to thank the authors for submitting an interesting and timely manuscript for review. I found it interesting to learn about the differences in awareness in this specific population and I think this article will be of interest to the journal's general readership. I did have the following comments/suggestions:
- Introduction: In general I found the Introduction informative however it is missing some perspective on the actual burden of ASD in Saudi Arabia (ie: incidence/prevalence if known). I also think the introduction would benefit from more information on Saudi Arabia itself - for example the general health literacy of citizens, what kinds of social media platforms are most utilized etc. If there is a low rate of ASD diagnoses in the kingdom for example, then the study would have much lower significance. In addition to this, what percentage of the population has access to social media platforms (if known)? Obviously, if only a well-educated portion of the population use these, this will skew data accordingly.
- Thank you for this valuable comment. We have now added information about the prevalence of ASD in Saudi Arabia and the Gulf countries in general, lines 112-117. Further information related to what kinds of social media platforms are most utilized is highlighted in the discussion section, lines 251-257.
- Methods: I would suggest listing ALL platforms on which the survey was advertised if possible. Also was the survey available in English, Arabic, both or any additional languages? Once again if only available in a single language, this will also skew data towards those with only a working knowledge of that language.
- Thank you for this valuable comment. We already mentioned all platforms used in line 138, which are Facebook and WhatsApp. We have now highlighted that the questionnaire tool was written in Arabic to ensure that the majority of Saudi Arabia's population could understand it, lines 138-140.
- Methods: My main concerns with the article are with the methods - specifically the questionnaire itself. Was the questionnaire validated in any way? Did the authors utilize a pilot group first before releasing the final version to the public? Also on Page 3 Line 105 the authors mention that the tool was developed based on "literature review". If that is the case, the authors should cite some of the pertinent literature (particularly if the current questionnaire did not undergo a rigorous validation process). If the survey/questionnaire was not validated, that should be mentioned as a limitation in the discussion.
- Thank you for this valuable comment. Yes, the clarity of the questionnaire items and whether they are measuring people's knowledge of ASD were discussed with two professional clinicians. They validated this and gave comments on some of the wording used in presenting the questionnaire items in order to make them more understandable to the general audience. Before disseminating the final questionnaire to the research population, all comments were addressed. Besides, a pilot study was conducted to confirm the questionnaire's clarity and comprehensibility. All these details is now mentioned in lines 147-155.
- We have now cited references used to develop the questionnaire tool, line138. We have now highlighted in the limitation section, that our questionnaire tool was not validated further, lines 338-339.
- Discussion: In general, I found the discussion lacking in terms of conclusions from the authors. On Page 8 Line 192 for example, the authors comment on social media being the primary source of participant knowledge - yet the authors do not attempt to explain why this may be so in Saudi Arabia specifically. What is it about that population that may make this so?
- Thank you for this valuable comment. We have now discussed the use of internet and social media in Saudi Arabia and these findings further in the discussion, lines 251-276.
- Discussion: As well on Page 8 in Paragraphs 3 and 4 the authors spend a lot of time discussing previous studies but do not give any indications on what their study may or may not tell us about the population of Saudi Arabia as they see ASD. Only the final Paragraph 5 suggests a few interventions but these are on briefly mentioned. There is also no mention of the limitations of the current study or how it may be improved in the future. As mentioned, the discussion needs to be modified to really address what the authors have gleaned from their study.
- Thank you for this valuable comment. We have now re-wrote our discussion to address the reviewer comment. Besides, we added the study limitations, implications and recommendation for future research, pages 8-10.
Reviewer 3 Report
This is an interesting cross-sectional study carried out between 10
June and September 2021 in Saudi Arabia, by means of an online questionnaire tool on 500 adult people, in order to assess knowledge about autism spectrum disorder (ASD) in terms of its etiology, patient features, abilities and needs. The results show that knowledge about ASD
in Saudi Arabia is moderate, and that females, persons with a master's degree, and those working in the healthcare field had a higher likelihood of knowing more about ASD.
In the Introduction (rows 70-74) the paragraph "Attention-deficit hyperactivity disorder (ADHD)...a separate genetic condition" should be deleted, because it is misleading. In fact, attention-deficit hyperactivity disorder (ADHD), social anxiety disorder, intellectual disability, and epilepsy are not all examples of ASDs, but they are comorbidities of ASD. Furthermore, ADHD is not the most frequent ASD disorder, but another neurodevelopmental disorder, according to DSM 5! The Authors quote fragile X syndrome as "...another example of a separate genetic condition", but is not the most representative example of a cause of ASD, since it often is not associated with ASD.
The questionnaire used by the Authors in this study should be added as supplementary material.
This questionnaire is not standardized, and this is a limitation of this study. The Authors should address this issue in the Discussion. There are also other methodological bias, correlated with the distribution using social media platforms (WhatsApp and Facebook). The Authors should also discuss this point.
Author Response
This is an interesting cross-sectional study carried out between 10
June and September 2021 in Saudi Arabia, by means of an online questionnaire tool on 500 adult people, in order to assess knowledge about autism spectrum disorder (ASD) in terms of its etiology, patient features, abilities and needs. The results show that knowledge about ASD
in Saudi Arabia is moderate, and that females, persons with a master's degree, and those working in the healthcare field had a higher likelihood of knowing more about ASD.
In the Introduction (rows 70-74) the paragraph "Attention-deficit hyperactivity disorder (ADHD)...a separate genetic condition" should be deleted, because it is misleading. In fact, attention-deficit hyperactivity disorder (ADHD), social anxiety disorder, intellectual disability, and epilepsy are not all examples of ASDs, but they are comorbidities of ASD. Furthermore, ADHD is not the most frequent ASD disorder, but another neurodevelopmental disorder, according to DSM 5! The Authors quote fragile X syndrome as "...another example of a separate genetic condition", but is not the most representative example of a cause of ASD, since it often is not associated with ASD.
- Thank you for this valuable comment. We have now corrected the above mentioned sentence as the following “Attention-deficit hyperactivity disorder (ADHD), social anxiety disorder, intellectual disability, and epilepsy are all examples of developmental disabilities that can coexist with concomitant mental diseases and a separate genetic disorder”. We have now corrected the second sentence as the following “ADHD is another neurodevelopmental disease that frequently coexists with ASD. It affects both children and adults and can lead to depression and anxiety”, lines 91-98.
The questionnaire used by the Authors in this study should be added as supplementary material.
- We have now addressed the above mentioned comment and added the questionnaire to supplementary material.
This questionnaire is not standardized, and this is a limitation of this study. The Authors should address this issue in the Discussion. There are also other methodological bias, correlated with the distribution using social media platforms (WhatsApp and Facebook). The Authors should also discuss this point.
- Thank you for this valuable comment. We have now highlighted these two points in the limitations, lines 331-340.
Reviewer 4 Report
- Page 1, lines 36-39: this sentence should be corrected: "slight condition" -?- At least, authors could introduce the three sub-categories of classical autism. But there is no "slight". The same for "not require assistance".
- Page 2, lines 46-49: check the phrases.
- Page 2, lines 54-55: there is no verb.
- Replace reference #21.
- On the 500 participants, how many had an autistic child in their family?
- Discussion. Key findning #2: authors could add more on this important finding, since this source of information could be also a source of mis-information.
- References #2 and #3 are the same.
Author Response
- Page 1, lines 36-39: this sentence should be corrected: "slight condition" -?- At least, authors could introduce the three sub-categories of classical autism. But there is no "slight". The same for "not require assistance".
- - Thank you for this valuable comment. We have now addressed the reviewer comment and rephrased our sentence, lines 38-42.
- Page 2, lines 46-49: check the phrases.
- Thank you for this valuable comment. We have now addressed the reviewer comment and rephrased our sentences, lines 54-66.
- Page 2, lines 54-55: there is no verb.
- Thank you for this valuable comment. We have now addressed the reviewer comment and rephrased our sentence and clarified it further, lines 67-76.
- Replace reference #21.
- - We have now corrected this reference as per the reviewer comment.
- On the 500 participants, how many had an autistic child in their family?
- - Unfortunately, we did not ask the participants whether they have autistic child in their family.
- Key findning #2: authors could add more on this important finding, since this source of information could be also a source of mis-information.
- - We have now addressed the reviewer comment in the discussion, lines 251-275.
- References #2 and #3 are the same.
- - We have now deleted the duplicate references.
Round 2
Reviewer 1 Report
change wording of "autistic disease" on p. 2, line 46 to "autism"
Reviewer 2 Report
I would once again like to thank the authors for their submission and careful addressing of my comments. The changes made have greatly enhanced the paper and I found the manuscript much more engaging and thoughtful. There are only minor spelling grammatical errors but I have no other comments. Great job!
Reviewer 3 Report
The Authors took in count my comments and queries, and so I think that this paper is now acceptable for publication.
Reviewer 4 Report
Authors well answered to my comments.